# Parcels v0.9: prototyping a Lagrangian Ocean Analysis framework for the petascale age

Michael Lange[1] and Erik van Sebille[2,3]

[1]Grantham Institute & Department of Earth Science and Engineering, Imperial College London, UK
[2]Institute for Marine and Atmospheric research Utrecht, Utrecht University, Utrecht, Netherlands
[3]Grantham Institute & Department of Physics, Imperial College London, UK

*Correspondence to:* Erik van Sebille (e.vansebille@uu.nl)

**Abstract.**

As Ocean General Circulation Models (OGCMs) move into the petascale age, where the output of single simulations exceeds petabytes of storage space, tools to analyse the output of these models will need to scale up too. Lagrangian Ocean Analysis, where virtual particles are tracked through hydrodynamic fields, is an increasingly popular way to analyse OGCM output, by mapping pathways and connectivity of biotic and abiotic particulates. However, the current software stack of Lagrangian Ocean Analysis codes is not dynamic enough to cope with the increasing complexity, scale and need for customisation of use-cases. Furthermore, most community codes are developed for stand-alone use, making it a nontrivial task to integrate virtual particles at runtime of the OGCM. Here, we introduce the new Parcels code, which was designed from the ground up to be sufficiently scalable to cope with petascale computing. We highlight its API design that combines flexibility and customisation with the ability to optimise for HPC workflows, following the paradigm of domain-specific languages. Parcels is primarily written in Python, utilising the wide range of tools available in the scientific Python ecosystem, while generating low-level C-code and using Just-In-Time compilation for performance-critical computation. We show a worked-out example of its API, and validate the accuracy of the code against seven idealised test cases. This version 0.9 of Parcels is focussed on laying out the API, with future work concentrating on support for curvilinear grids, optimisation, efficiency and at-runtime coupling with OGCMs.

## 1   Introduction

Lagrangian Ocean Analysis, where virtual particles are tracked within the flow field of hydrodynamic models, has over the last two decades increasingly been used by physical oceanographers and marine biologists alike (Van Sebille et al., submitted).The particles can represent passive parcels of seawater (e.g. Döös, 1995; Blanke and Raynaud, 1997) or its constituent tracers such as nutrients (e.g. Jönsson et al., 2011; Qin et al., 2016), as well as particulate matter such as microbes (e.g. Hellweger et al., 2014; Doblin and van Sebille, 2016), larvae (e.g. Cowen et al., 2006; Paris et al., 2005; Teske et al., 2015; Cetina-Heredia et al., 2015), pumice (e.g. Jutzeler et al., 2014), plastic litter (e.g. Lebreton et al., 2012), or icebergs (e.g. Marsh et al., 2015). The trajectories of the virtual particles can be used to analyse the flow within Ocean General Circulation Models (OGCMs) and other velocity fields for dispersion characteristics (e.g. Beron-Vera and LaCasce, 2016), Lagrangian Coherent Structures (e.g. Haller, 2015), water mass pathways and transit times (e.g. Rühs et al., 2013), Lagrangian streamfunctions (e.g. Döös et al.,

2008) and biological connectivity between regions (e.g. Kool et al., 2013). See Van Sebille et al. (submitted) for an extensive review on Lagrangian Ocean Analysis.

There are currently three main community codes available to calculate the trajectories of virtual particles in Ocean General Circulation Models: Ariane (Blanke and Raynaud, 1997), TRACMASS (Döös et al., 2013; Döös et al., 2017), and the Connec-
tivity Modeling System (CMS, Paris et al., 2013). These codes, being open-source and having excellent support teams, have served the wider community very well over the past decades. However, it is not clear that these three codes will be able to scale up easily to the petascale age of computing, where particle trajectory codes will need to be able to deal with potentially petabytes of hydrodynamic field data and gigabytes of particle trajectory data. Exploring advanced optimization strategies to overcome these big-data challenges, such as coupled (online) execution with the host OGCM or reducing the volume of hydro-
dynamic data by selectively filtering data regions based on particle locations, will require a flexible execution model that can dynamically be adapted to complement the respective data and execution formats of various host OGCMs.

Furthermore, the current stack of codes is mostly built for the tracking of water parcels or passive particulates. While the CMS and TRACMASS do support the addition of diffusive processes through Markovian stochastic models (e.g. Griffa, 1996), it is non-trivial to incorporate 'behaviour' of particulates to these codes. Effortless incorporation of behaviour such as sinking,
fragmentation, or even swimming to particulates would simplify exploration of the dynamics of particulates such as fish, icebergs and marine debris.

Here, we describe a novel framework for computing Lagrangian particle trajectories, named Parcels ('Probably A Really Computationally Efficient Lagrangian Simulator'). Being developed from the ground up with scalability and performance in mind, we hope that this Parcels framework will be able to keep up with OGCM development for the coming decades,
particularly by being scalable and efficient at reading in hydrodynamic data. We have furthermore focussed on flexibility and customisability of the particle dispersion schemes, so that it is relatively straightforward to add new functionality such as active particle behaviours.

We have decided to brand this version of Parcels as v0.9, signalling that while in principle it is feature-complete, the code is not nearly as fast and efficient as we envision it to be in the future. Improving performance will be the main priority as we
work towards v1.0. We invite all interested researchers to contribute to the development by starting to use the code.

While development efforts of Parcels focus on oceanographic applications, the Parcels framework should in principle also be adaptable to atmospheric particle tracking simulations. Models such as FLEXPART (Stohl and James, 2005) and the MetOffice NAME model (Jones et al., 2007) are state-of-the-art and have an excellent track-record in the field of atmospheric dispersion modelling, but perhaps some of the ideas presented here could be incorporated or used in these models too.

This paper is structured as follows: in the next section, we will describe the philosophy behind the Parcels code. We then present a worked-out example of an application of Parcels for an actual scientific experiment in Sect. 3. Following that, we evaluate the accuracy of the code in Sect. 4, by comparison to analytical solutions in idealised test. We provide a future outlook in Sect. 5, before concluding in Sect. 6.

## 2 Prototype design and philosophy

A key contribution of the new Parcels v0.9 framework is to define a set of interfaces and composable abstractions that encapsulate the various processes required to create dynamic and extensible Lagrangian models that feature direct interactions between particles and an associated OGCM grid. The design follows modern scientific software engineering practices, providing high levels of modularity and flexibility with a clear intent to further specialize various sub-components at a later stage. The interfaces provided in Parcels are therefore intended to capture the general domain-specific challenges posed by particle tracking for Lagrangian Ocean Analysis. The overall design philosophy, as well as the structure of the code, are driven by three major design considerations:

- **Extensibility** – While the core algorithm of Lagrangian particle models is concerned with the advection and dispersion of passive particles that constitute infinitely small point parcels, practical oceanographic applications often require more complex behaviour of the particles. Potential extensions towards individual-based modelling of particulates to simulate biological species or marine debris will require extensions to particle data definitions and programmable behavioural customisation at a per-particle level.

- **Compatibility** – Particle tracking in oceanography requires the close coupling of computational particles to velocity data that defines the hydrodynamic flow field. Parcels aims to make as little assumptions about the nature and structure of the hydrodynamic fields as possible, so as to be compatible with various types of OGCMs and data formats. While the focus in this v0.9 is on utilising offline data, this includes considerations for at-runtime coupling with OGCMs in the future.

- **Dynamic data** – Particle data is sparse in nature and can, depending on application context, exhibit very dynamic data access patterns where new particles are inserted and deleted from the active set at runtime. For this reason, structured compile-time performance optimisations and parallelisation strategies are insufficient, and Just-In-Time scheduling is required to handle the amorphous data parallelism inherent in dynamic particle applications (Pingali et al., 2011).

The above list of requirements suggests that a static compile-time approach is likely to provide insufficient flexibility to adjust to the various scientific contexts in which oceanographic particle tracking might be utilised. For this reason Parcels is based on the domain-specific languages paradigm, which aims to decouple the problem definition as defined by the scientific modeller from the implementation that is ultimately executed on a particular hardware architecture. This approach is based on automated code generation at runtime and creates a separation of concerns between domain scientists and computational experts that allows hardware-specific performance optimisation and thus greater flexibility with respect to advances in high-performance computing resources.

Since the prototype of the Parcels framework presented here provides a conceptual blueprint for future versions, we define a clear set of abstractions for the following three software layers:

- **User-facing API** – The primary objective of Parcels is to provide a user-friendly, clear and concise API for scientists to perform oceanographic particle tracking experiments with very little effort, while leaving room for customisations that

go beyond traditional configuration files. For this reason Parcels provides a high-level Python API that enables users to define a complete model in a small number of lines of code (see examples in Sect. 3). For more advanced models, the API also provides enough scope to fully control the variable layout of particles in memory, as well as to define custom behaviour via individual kernel operations.

5     – **Execution layer** – The transient nature of Lagrangian particles implies that many practical oceanographic applications rely on particle sets that may grow and shrink dynamically, while also relying on external hydrodynamic field data that might be sampled at a timestep much different from the primary particle loop. This complex parameter variability entails that the core loop that updates individual particle states needs to be highly dynamic and flexible, as well as highly optimised for large-scale applications. Parcels aims to encapsulate the core parameters of the particle update loop so 10     as to establish an interface for integration with a variety of external host OGCMs, and leaves enough scope for more advanced performance optimisations in the future.

    – **Data layout** – The two fundamental types of data involved in Lagrangian particle tracking algorithms constitute field data provided by the external OGCM, as well as data on the particle state. Since the data layout for particle data might change with future performance optimisations, and the memory layout of field data depends on the OGCM implementation, 15     Parcels provides high-level abstractions for both types of data, allowing the actual data layout in memory to change.

The abstractions shown in Fig. 1 comprise the core functionalities provided by the framework. The primary input in the user layer consists of generic definitions of the particle variables for individual types of particles, alongside an interface to define the computation kernels. Parcels' core execution loop uses this information to update particle data given external parameters, such as timestepping constraints, and interpolated hydrodynamic field data. Thus, given a stable user-level API and a highly 20 modular code structure, it is possible to implement various applications and experiments without commiting to a particular implementation, while leaving enough scope for further development and future performance optimisation 'under the hood'.

## 2.1   Programmable user interface

The prototype presented in this paper provides a highly flexible user API that allows users to define complete models via the Python programming language. The user hereby manages creation, execution and customisation of individual sets of particles, 25 as well as combinations of computational kernels to update the particle state. In contrast to traditional configuration files, this approach provides the user with native compatability with the open-source libraries and tools available in the scientific Python ecosystem.

The key components of Parcels' overall class structure are depicted in Fig. 2. The definition of the variables that constitute a single particle is hereby encapsulated in the `Particle` class, while container objects of type `ParticleSet` provide the 30 runtime handling and management of particle data. Python *descriptor* objects are used to generically define the compound data type underlying each type of particle, leaving allocation and memory layout choices to the particular implementation of the data container structure.

The computational behaviour of particles is encapsulated through the `Kernel`. Parcels provides a set of pre-defined advection methods, as well as allowing users to define custom behaviour programmatically. Multiple kernels can be concatenated, allowing users to incrementally build complex behaviour from individual components.

### 2.1.1 Advection algorithm

At its core, computing Lagrangian particle trajectories is equivalent to solving the following equation:

$$\boldsymbol{X}(t + \Delta t) = \boldsymbol{X}(t) + \int_{t}^{t+\Delta t} \boldsymbol{v}(\boldsymbol{x}, \tau) \, \mathrm{d}\tau + \Delta \boldsymbol{X}_b(t), \tag{1}$$

where $\boldsymbol{X}$ is the three-dimensional position of a particle, $\boldsymbol{v}(\boldsymbol{x}, t)$ is the three-dimensional velocity field at that location from an OGCM, and $\Delta \boldsymbol{X}_b(t)$ is a change in position due to 'behaviour'. The latter can itself be an integration of a (three-dimensional) velocity field, for example when a particle sinks downward because of a negative buoyancy force.

In Parcels, the trajectory equation (1) is by default time-stepped using a $4^{th}$ order Runge-Kutta scheme, although schemes for Euler-Forward and adaptive Runge-Kutta-Fehlberg integration (RKF45, e.g. Alexander, 1990) are also provided. In principle, the Parcels framework should be flexible enough to also implement integration using the discrete analytical streamtube method (Blanke and Raynaud, 1997; Döös et al., 2017).

### 2.1.2 Custom kernels

Lagrangian particle tracking in the ocean often involves more complex displacement schemes than simple velocity-driven advection. For example, in the presence of turbulence, a Random Walk kernel or Brownian motion is required, while ocean ecology models often include active locomotion. Parcels therefore allows users to create generic kernel functions by providing native Python functions that adhere to the function signature `KernelName(particle, fieldset, time, dt)`. Within these kernel functions, users can access built-in particle state variables, such as `particle.lat` and `particle.lon`, or

user-defined ones. Access to field data from within kernels is provided through the `fieldset` object, which provides fields as named properties, for example `fieldset.U` for the zonal velocity. Interpolation of field data is implemented via overloaded member access on the field object (square bracket notation), allowing user to express field sampling as `fieldset.fieldname[time, lon, lat, depth]`.

In addition to kernels that update the internal state of particles, Parcels' execution engine also enables users to customize

the behaviour of particles under various error conditions. For this, a similar type of kernel function can be created and passed to the execution call, mapped to a particular error type that might be triggered during the main particle update, for example `OutOfBoundsError`.

## 2.2 Execution and JIT compilation

The update of the internal state of particles is facilitated by a dynamic loop, which applies a user-defined combination of kernels to each particle in a `ParticleSet`. The primary particle update loop can either be run with a forward timestepping, or in a time-backward mode to enable inverse modelling. For this central update loop, Parcels provides two modes of execution:

- **Scipy mode:** A pure Python mode that utilises `interpolator` objects provided by the Scientific Python package (SciPy) to perform interpolation of field data. This mode is primarily intended as a debug option due to the performance penalty of running kernels in the Python interpreter itself.

- **JIT mode:** Runtime code generation and Just-In-Time compilation (JIT) are utilised to generate low-level C code that performs the particle state update and field data interpolation. The code generation engine hereby primarily translates
a restricted subset of the Python language into equivalent C code, while a set of utility modules provides auxiliary functionality such as random number generation or mathematical utilities (`math.h`).

The execution mode of the particle update loop is determined by the type of the particle (`ScipyParticle` or `JITParticle`) used to create the `ParticleSet`. Development of new features in the current Parcels prototype is strongly driven by the fact that both modes are intended to be semantically equivalent. This means that new features can rapidly be developed using the
full flexibility of the Python interpreter, providing a template implementation and test case for implementation in the computationally more efficient JIT mode.

Parcels' dynamic update loop also provides an `interval` keyword to impose a secondary sub-timestepping that allows for direct coupling with a host OGCM in the future. The dynamic composition of multiple timestepping intervals might also be used for future data and performance optimisation strategies, for example directed prefetching of regional field data. Such
strategies, as well as a potentially more intricate execution engine, have to be explored carefully to successfully tackle the big-data challenges facing Lagrangian tracking codes in the petascale age.

## 2.3 Interpolation

The interaction of particles with their enclosing fields is currently limited to interpolating field data onto the current particle position. In the SciPy debug mode this is facilitated by `scipy.interpolate.RegularGridInterpolator` objects
and supports linear and nearest-neighbour interpolation. Equivalent low-level C routines are also included in the Parcels source code as macros that can be inlined into the generated C kernel code by the code generation engine. More advanced interpolation methods, such as quadratic, cubic or spline interpolation, may easily be added in future releases if a fast C implementation can be provided with Parcels' internal header files.

One of key performance advantages of using runtime code generation is the ability to inline bespoke grid interpolation
methods with the user-defined kernels in Parcels to avoid the Python interpreter overhead of repeatedly calling native Python interpolation functions. This overhead can be quite significant due to the high frequency at which the associated field data needs to be sampled. This can be illustrated using the "Steady-state flow around a peninsula" test case discussed in Sect. 4.2.4, where

100 particles are advected for 20 hours with a timestep size of 30 seconds. While the sequential execution time of the pure Python implementation runs in 305.92 seconds, the auto-generated JIT kernels can run the same experiment in 1.74 seconds, a speedup of over 150×.

## 2.4 External field data

Parcels v0.9 supports external field data from NetCDF files, with a configurable interface to describe the input data and variable structure. The data is encapsulated in individual `Field` objects, which are accessible from within particle kernels via provided interpolation routines. Individual fields are stored in a `FieldSet` container class, which may also provide global meta-data to the kernel execution engine at runtime.

Currently, only linear interpolation schemes are implemented in Parcels, both in space and in time. In space, Parcels can currently only work on regular grids (i.e. where the grid dimensions are functions of only longitude, only latitude or only depth). However, support for unstructured grids is a priority for the next release of the code, Parcels v1.0.

## 3   A worked-out example: tracking virtual foraminifera in the Agulhas region

To highlight some of the prototype design and philosophies of the Parcels API, we here present a worked-out example code of a previously-published scientific experiment. This example follows the experimental design of Van Sebille et al. (2015), where the goal was to investigate the temperatures that planktic foraminifera experience during their lifespan as they drift with the currents in the upper ocean. In particular, that study looked at the variability of lifespan-averaged temperatures of foraminifera that all end up on one single location on the ocean floor (e.g. Peeters et al., 2004; Katz et al., 2010).

Figure 1b of Van Sebille et al. (2015) depicted the origin of virtual planktic foraminifera that end up on a site just off the coast of Cape Town (17.3°E, 34.7°S), at 2,440 meter water depth. The virtual particles were released at that site and then tracked in time-backward mode. There were two phases to the experiment: in the sinking phase, the foraminifera were tracked back as they sunk at 200 meter per day to the ocean floor, while being advected by the (deep) ocean circulation. In the lifespan phase, the particles were then tracked further backward in time as they were advected by the horizontal circulation at their 50m dwelling depth. During this last phase, temperature along their trajectory was recorded at daily interval.

While the original experiment was computed with the Connectivity Modelling System (Paris et al., 2013), here we have re-coded it using the Parcels API. This experiment setup is a fitting one, as it combines a number of the API highlights of Parcels: custom kernels, NetCDF I/O, and field sampling. The full Python code for this experiment in Parcels is available at https://doi.org/10.5281/zenodo.823994. Below, we emphasise some of the key statements in the Python script.

## 3.1   Reading the FieldSet

The hydrodynamic fields that carry the foraminifera come from the OFES model (Masumoto et al., 2004) and can be accessed from http://apdrc.soest.hawaii.edu/datadoc/ofes/ncep_0.1_global_3day.php. Three-dimensional velocities and temperature are available on 1/10° horizontal resolution, on 54 vertical levels, and are stored as three-day averages. The bash script

`get_ofesdata_agulhas.sh` provided at https://doi.org/10.5281/zenodo.823994 was used to download snapshot numbers 3165 to 3289, covering the year 2006, in a subdomain around the core site off Cape Town (note, the total file size is 6GB).

While the 6GB file size for this example is not excessively large and could in principle be loaded into memory all at once, this will not be possible for `FieldSets` with larger regional domains or longer time series. Hence, Parcels provides a system to read in hydrodynamic fields during particle integration, at any time storing only three consecutive timeslices (e.g. Paris et al., 2013). See also Section 3.4.

After the first three days of hydrodynamic fields are read in through a call to the user-defined `set_ofes_fieldset` function (see the `example_corefootprintparticles.py` script for the exact formulation of this function, which requires as input a set with filenames, provided as a list of arbitrary length), three global constants are added to the `FieldSet`

```
fieldset.add_constant('dwellingdepth', 50.)
fieldset.add_constant('sinkspeed', 200./86400)
fieldset.add_constant('maxage', 30.*86400)
```

These constants will be used later in the custom kernels controlling the movement of the particles.

## 3.2 Defining the ParticleSet

Apart from information on their location and time, the virtual foraminifera particles will need two extra `Variables`: the sea water temperature at their present location, and their age. Therefore, we define a new particle class, which inherits from the standard `JITParticle`:

```
class ForamParticle(JITParticle):
    temp = Variable('temp', dtype=np.float32, initial=np.nan)
    age = Variable('age', dtype=np.float32, initial=0.)
```

And we then define a `ParticleSet` containing a single particle as

```
pset = ParticleSet(fieldset=fieldset, pclass=ForamParticle, lon=[17.3], lat=[-34.7],
                   depth=[2440], time=fieldset.U.time[-1])
```

## 3.3 Defining the custom kernels

We need to define four custom kernels: one that causes the particle to sink after it dies, one that keeps track of its age and deletes it once it reaches its maximum age, one that samples the temperature at its location, and one that deletes the particle when it reaches a boundary of the domain (since we only have hydrodynamic data in a subset of the global OFES domain). Note that while in principle the first three could be written in one Kernel, here we write three separate kernels and then concatenate these with the built-in `AdvectionRK4_3D` kernel.

The first kernel, controlling the sinking of the particle after it died (i.e. the first twelve days in our reverse-time experiment), can be written as

```
def Sink(particle, fieldset, time, dt):
    if particle.depth > fieldset.dwellingdepth:
        particle.depth = particle.depth + fieldset.sinkspeed * dt
    else:
        particle.depth = fieldset.dwellingdepth
```

The second kernel, which keeps track of the age and deletes the particle when it reaches `maxage`, can be written as

```python
def Age(particle, fieldset, time, dt):
    if particle.depth <= fieldset.dwellingdepth:
        particle.age = particle.age + math.fabs(dt)
    if particle.age > fieldset.maxage:
        particle.delete()
```

The third kernel, which samples the temperature, can be written as

```python
def SampleTemp(particle, fieldset, time, dt):
    particle.temp = fieldset.temp[time, particle.lon, particle.lat, particle.depth]
```

These three kernels are then concatenated with the `AdvectionRK4_3D` kernel as

```python
kernels = pset.Kernel(AdvectionRK4_3D) + Sink + SampleTemp + Age
```

Where at least one of the kernels needs to be cast into a `Kernel` object for the overloading of the + operator as a kernel concatenator to work.

Finally, the kernel that deletes a particle if it reaches one of the lateral boundaries and which will be invoked through the

error recovery execution is

```python
def DeleteParticle(particle, fieldset, time, dt):
    particle.delete()
```

## 3.4   Executing the particle set

The `ParticleSet` can now be integrated with a call to `pset.execute()`. This method requires as input the list of kernels, the starttime of the execution loop, the runtime of the execution loop, the Runge Kutta integration timestep (here taken to be

5 minutes), the interval at which output is written (here once per day), and the recovery kernel that gets called when a Particle crosses the boundary of the regional domain.

As mentioned in section 3.1, only three timeslices are held in memory at any one time. The loading of new fields is controlled by the `fieldset.advancetime()` method, which replaces the oldest timeslice with a new one (held in this case in `[snapshots[s]]`). This also means that the executing of the ParticleSet has to be done within a loop:

```python
for s in range(len(snapshots)-5, -1, -1):
    pset.execute(kernels, starttime=pset[0].time, runtime=delta(days=3),
                 dt=delta(minutes=-5), interval=delta(days=-1),
                 recovery={ErrorCode.ErrorOutOfBounds: DeleteParticle})
    fieldset.advancetime(set_ofes_fieldset([snapshots[s]]))
```

There is another reason to call the `pset.execute` method within a loop: it allows for a new particle to be released every three days (the frequency with which hydrodynamic data is available). This happens within the for-loop through a call to

```python
pset.add(ForamParticle(lon=[17.3], lat=[-34.7], depth=[2440], fieldset=fieldset))
```

## 3.5   Saving and plotting the output

The Parcels framework allows for storing of the locations of the particle to disk on-the-fly in NetCDF files, following the Discrete Sampling Geometries section of http://cfconventions.org/cf-conventions/v1.6.0/cf-conventions.html\#discrete-sampling-geometries,

and is hence CF-1.6-compliant. Storing of the particle trajectories and properties such as age and along-track temperature happens in the for-loop through calls to

```
pfile.write(pset, pset[0].time)
```

Since particles are continually added to and deleted from the ParticleSet, the ParticleFile needs to be stored in 'indexed' format, where for each variable all particle states are written in one long vector.

```
pfile = ParticleFile(outfile, pset, type="indexed")
```

These long vectors in Indexed format, however, are not very easy to work with, so Parcels provides the utility script convert_IndexedOutputToArray to convert an Indexed NetCDF file to array format.

The particle trajectories can then be plotted using the matplotlib and Basemap libraries, see Fig. 3. This figure shows the temperature recorded on each day during the lifespan of all virtual particles. It highlights that foraminifera that end up on the ocean floor off Cape Town travel hundreds to thousands of kilometers during their lifespan, and that while some originate from the Agulhas Current as far north as 27°S, others originate from the much colder Southern Ocean south of 40°S.

## 4 Model evaluation

Evaluation of a code-base's accuracy and performance is a key component of its validation and roll-out. For this Parcels v0.9, performance and speed are not a priority; these will be the focus for the v1.0 release (see also Sect. 5). Instead, while developing Parcels v0.9 we have concentrated on accuracy.

### 4.1 Unit tests and continuous integration

Following best practices in software engineering, we have incorporated Unit Testing and Continuous Integration into the development cycle of Parcels. Every push of code changes to github automatically triggers a validation of the entire code base (an important component of the Continuous Integration paradigm), through the travis-ci.org web service.

The validation of the code base is done through so-called unit tests; small snippets of code that test individual components of the codebase. Parcels v0.9 has over 150 of these unit tests, which check the integrity and consistency of the codebase. Where relevant, these unit tests are run in both Scipy and JIT mode, to test both modes of executing the kernels.

The following Python snippet shows a typical example of a unit test for Parcels (as included in the test_particle_sets.py file). It performs the test that Particles in a ParticleSet indeed get their assigned longitudes and latitudes. While this may seem a trivial test, these kinds of unit tests can help prevent bugs.

```
@pytest.mark.parametrize('mode', ['scipy', 'jit'])
def test_pset_create_lon_lat(fieldset, mode, npart=100):
    lon = np.linspace(0, 1, npart, dtype=np.float32)
    lat = np.linspace(1, 0, npart, dtype=np.float32)
    pset = ParticleSet(fieldset, lon=lon, lat=lat, pclass=ptype[mode])
    assert np.allclose([p.lon for p in pset], lon, rtol=1e-12)
    assert np.allclose([p.lat for p in pset], lat, rtol=1e-12)
```

Ideally, the full set of unit tests means that no change of the code can ever break another part of the code, since some of the unit tests would then fail. Of course, in reality the completeness of the unit tests can never be guaranteed, but during Parcels development we have attempted to provide unit tests for a broad spectrum of the Parcels functionality and code.

## 4.2 Idealised and analytic test cases

Following the list of standard tests of particle tools, as described in Sec. 6 of Van Sebille et al. (submitted), we have validated the accuracy of Parcels v0.9 against seven idealised and analytical test cases. In this section we will describe the results in detail. All test cases are run with Runge-Kutta4 integration and in JIT mode. In each case, the hydrodynamic velocities are generated within the Python scripts and converted directly to a `FieldSet` (i.e. without first storing these fields in NetCDF format).The Python code for all testcases is available at https://doi.org/10.5281/zenodo.823994.

### 4.2.1 Radial rotation with known period

The first test case is that of a simple counter-clockwise solid-body rotation with a period of 24 hours. Velocities are defined on a $(20 \times 20)$ km Arakawa A-grid centered at the origin with a 100 m horizontal resolution. Solid-body radial velocities $(u, v) = (-\omega r \sin(\phi), \omega r \cos(\phi))$, with $r$ and $\phi$ the radius and angle from the origin and $\omega = 2\pi/86,400$ s the angular frequency, are then computed on that grid.

Four particles are started at $x = 0$ km and $y = (1000, 2000, 3000, 4000)$ km and then advected for 24 hours, using an RK4 timestep of 5 minutes, and with particle positions stored every hour (Fig. 4a). All four particles indeed follow the flow for the full circle. The maximum distance error after this 24 hours advection is less than 3mm, on path lengths of more than 5km.

### 4.2.2 Longitudinal shear flow

The second test case tests the ability of the Parcels code to convert between spherical longitude/latitude space and local flat Euclidian space. When defining a `FieldSet` on a spherical mesh, Parcels automatically performs this conversion under the hood. To test its accuracy, an idealised flow on a sphere at $1°$ horizontal resolution is created, with a uniform zonal velocity of 1 m/s and no meridional velocity. A total of 31 particles are then released on a north-south line, with a meridional spacing of $3°$. These particles are advected for 57 days, using an RK4 timestep of 5 minutes and output saved every day (Fig. 4b). The main panel shows trajectories in planar projection, with the inset showing the same trajectories in orthographic projection.

At a speed of 1 m/s, the particles travel $4.9 \cdot 10^6$ m in the 57 days. At the equator, this amounts to almost $45°$ of longitude, but because of the cosine-dependence of zonal distance with latitude, particles closer to the poles travel farther in degrees (main panel in Fig. 4b). The inset of Fig. 4b, nevertheless, shows that in an orthographic projection, all particles travel the same distance.

### 4.2.3 Advection due to a time-oscillating zonal flow

The third test case tests the ability of Parcels to cope with simple time-varying flow. The flow in this case is a uniform meridional flow of $v = A = 0.1$ m/s, and an oscillating zonal flow with $u(t) = A\cos(\omega t)$ where $\omega = 2\pi/T$ and the period is $T = 1$ day. The time resolution of the `FieldSet` is 5 minutes, and since the flow is constant in space there are only two grid cells in each of the horizontal directions. A total of 20 particles are then released on a zonal line at $y = 0$ km and advected for 4 days, using an RK4 timestep of 5 minutes and storing output every 3 hours (Fig. 4c).

The analytical flow for the paths of these particles is $y(t) = At$ and $x(t) = x_0 + A/\omega\sin(\omega t)$ where $\omega = 2\pi/T$ and $x_0$ is the zonal start location of the particle. Indeed, all particles follow these analytical pathways very closely (Fig. 4c), with largest positional errors after 4 days being 6 cm in the zonal direction and 4 mm in the meridional direction.

### 4.2.4 Steady-state flow around a peninsula

The test case of steady-state flow around a peninsula follows a description by Ådlandsvik et al. (2009) and was also used as a validation test case in the article describing the Connectivity Modeling System (Paris et al., 2013). Starting from the analytical expression for a streamfunction $\Psi$ of a steady-state flow around a peninsula, analytical expressions of the zonal and meridional component of velocity are solved on a $(1° \times 0.5°)$ Arakawa A-grid at $1/100°$ horizontal resolution. A set of 20 particles is seeded just off the western edge of the domain, and then advected with the flow for 24 hours using an RK4 timestep of 5 minutes and particle positions stored every hour (Fig. 4d, where the brown semi-circle is the peninsula).

Since the particles should follow streamlines, a comparison of the interpolated streamfunction value at $t = 24$ hours to that at $t = 0$ hours gives an estimate of the error. The largest error is 0.008 m$^2$/s, which corresponds to a positional error of $10^{-5}$ degrees, or 1 meter. Indeed, Fig. 4d shows that the particle trajectories closely follow the dashed streamlines.

### 4.2.5 Steady-state flow in a Stommel gyre and western boundary current

The test case of the Stommel gyre follows a description in Fabbroni (2009), and provides an analytical solution to the stream-function field of a Stommel gyre and western boundary current. Here, we compute the meridional and zonal central derivatives of this streamfunction field to generate zonal and meridional velocities, respectively, on a $(10,000 \times 10,000)$ km Arakawa A-grid at 50 km horizontal resolution. A set of four particles is seeded on a line crossing the western boundary, at $y = 5,000$ km, and then advected for 50 days with an RK4 timestep of 5 minutes and the particle positions stored every 24 hours (Fig. 4e).

Since the particles should follow streamlines, the deviation of particles from the streamlines is a measure of the accuracy of the method. Fig. 4e shows that all three particles stay close to their streamline throughout the 50 day advection period. The largest error is 0.05 m$^2$/s, which corresponds to a positional error of less than 5 km.

### 4.2.6 Damped inertial oscillation on a geostrophic flow

The test case of a damped inertial oscillation on a geostrophic flow follows Fabbroni (2009) and Döös et al. (2013). In this test case, the velocity varies over the entire domain, following an analytical time-dependent equation. Here, we use a time

resolution of 5 minutes for the velocity field. A particle is then seeded at the origin and advected for four days, with a RK4 timestep of 5 minutes and output stored every hour (Fig. 4f). After four days of advection, the positional error of the particle, as compared to the analytical solution, is less than 5 cm.

### 4.2.7 Brownian motion with uniform $K_h$

The test case of Brownian motion with uniform $K_h$ tests for the accuracy and implementation of the random number generator. Here, a total of 100,000 particles are seeded at the origin of a $(60 \times 60)$ km grid centered around the origin with zero velocities, and then diffused using a normal variate random number distribution with $K_h = 100$ m$^2$/s. The particles are diffused for 1 day with a timestep of 5 minutes (Fig. 4g). The two-dimensional normalised histogram agrees very well with the analytical solution of this Brownian motion: a two-dimensional Gaussian with a mean at the origin and standard deviation of $\sigma = \sqrt{2K_h \Delta t} = 4.16$ km.

## 5 Future outlook

As mentioned before, Parcels v0.9 is a prototype. The core contributions of this paper are both the API, as well as the design philosophy which enables a wide range of valuable future improvements of the framework. Below, we discuss some of the conceptual ideas for these planned improvements.

### 5.1 Performance optimisation

The primary performance optimisation in version 0.9 of Parcels is the automated generation of C kernel code to allow inlining of field evaluation routines. However, several future optimisations have been planned during the design of the code, based around considerations for irregular data processing. Since dynamic addition and deletion of particles is a common feature of many oceanographic use cases, no assumptions about data layout or iteration protocol have been made in the high-level API of particle sets, allowing more optimised implementations in the future. The use of dynamic code generation at runtime also enables further automated specialisation of kernel code, while allowing us to define a clear initial interface for kernel customisation.

In addition to optimising the execution of particle kernels, the extensive interaction with hydrodynamic field data constitutes a considerable cost of the overall computation - a cost that is likely to dominate overall execution if large sets of hydrodynamic field data are to be read from files. Multiple potential approaches can be considered in future versions of Lagrangian particle tracking codes:

- Directly coupled (online) runs within the host OGCM can completely avoid the bandwidth bottlenecks imposed by reading dense field data from disk, at the expense of additional computation. For simulations at local scales with a high particle density, this trade-off might prove beneficial, for example for regional studies on marine ecology.

- For global-scale models that require offline hydrodynamic field data but feature a low particle density with high localization, the total volume of data read from disk might be drastically reduced by explicitly prefetching local subsets of field data based on particle locations. Such a mechanism would require the use of additional geospatial indexing methods, for example via octrees or r-trees (Isaac et al., 2015; Schubert et al., 2013), that decompose the grid into individual sub-regions and provide fast indexing methods. Using explicit prefetch directives in the dynamic execution loop might also enable overlapping of asynchronous file reads with effective computation to further amortize file I/O overheads.

The modularity of Parcels' internal abstractions, as well as the composability of kernels and the flexibility provided by the dynamic execution loop should facilitate extensive experimentation and exploration with such advanced optimization techniques, without the need for users to change any high-level algorithmic definitions. The use of advanced data handling and task-scheduling libraries, such as Dask (Rocklin, 2015) or Xarray (Hoyer and Hamman, 2017), might also be utilised to quickly achieve efficient out-of-core data management in Parcels.

### 5.1.1 Towards parallellisation

The current version of Parcels is not in itself parallel due to two restrictions:

- The primary input format of field data in the v0.9 prototype is NetCDF-based field data, so that parallellisation requires an explicit domain decomposition and a parallel file reader. The current version of the `netcdf` Python package does not provide these features. Alternative implementations of the NetCDF file format, such as Xarray, might be leveraged in future versions of Parcels to provide parallel data management.

- Exchanging particle information between parallel processors is currently not supported, although it is deemed a critical feature for the next release (v1.0).

### 5.2 Community building and kernel sharing

One of the key ideas between the development of Parcels is for it to be a flexible and extendable codebase, where particle behaviour can easily be customised. The worked out example in Sect. 3 shows that many types of behaviour (sinking, aging, etc) can be coded in a few lines of Python code.

The customisability of Parcels enables a multitude of oceanographic modelling, from water parcels to plankton to plastic litter to fish. We therefore envision an active community of Parcels users who share and discuss kernel development. We encourage anyone who wishes to share their custom kernels to upload them onto github, and we will provide a properly referenced library of user-contributed kernels for others to reuse on oceanparcels.org.

### 5.3 Towards runtime integration with OGCMs

Although the current version of Parcels primarily uses off-line field data, the overall design of the particle exectuion engine is designed to be compatible with a variety of OGCMs for directly coupled (at-runtime) simulations. In particular, the current

`Field` interface can easily be extended to provide interpolation routines for various types of field data, for example based on unstrctured meshes, while the primary particle update loop provides a mechanism for host models to dictate a model timestep size that varies from that of the particle update. Moreover, the explicit generation of C code allows Parcels kernel code to be easily injected into existing ocean modelling frameworks, while the provision of error-recovery kernels can guarantee
progression of the coupled model.

## 6 Conclusions

Here, we have introduced a new framework for Lagrangian ocean analysis that focusses on customisability, flexibility and ease-of-use. This v0.9 of Parcels is very much a prototype, providing a proof-of-concept of the API and showcasing how it can be used to create high-level Python code for full-fledged scientific experiments. We also assess the accuracy of the
10 current implementation, with the idea to provide a benchmark for future versions. Future development will focus on increasing efficiency of the framework, and also towards providing easy tools to port the generated C-code of Parcels experiments to at-runtime integration within OGCMs.

## 7 Code availability

The code for Parcels is licensed under the MIT license and is available through github at github.com/OceanParcels/parcels.
The version 0.9 described here is archived at Zenodo at https://doi.org/10.5281/zenodo.823562. More information is available on the project webpage at oceanparcels.org.

*Author contributions.* ML and EvS developed the code and wrote the manuscript jointly.

*Competing interests.* The authors declare no competing interests

*Acknowledgements.* The initial ideas for the Parcels framework were the result of very fruitful discussions with the attendees of the "Future of
20 Lagrangian Ocean Modelling" workshop, held at Imperial College London, UK, in September 2015. Funding for this workshop was provided through an EPSRC Institutional Sponsorship grant to EvS under reference number EP/N50869X/1. EvS is supported through funding from the European Research Council (ERC) under the European Union's Horizon 2020 research and innovation programme (grant agreement No 715386). The OFES simulation was conducted on the Earth Simulator under the support of JAMSTEC. We thank Joe Scutt-Phillips, Ronan McAdam, Joel Kronberg, Thomas Stokes, Nathaniel Tarshish, Michael Hart-Davis, Birgit Sutzl, Ben Snowball, Samuel Wetherell and David
Ham for their support in testing and developing aspects of the Parcels code. We thank the reviewers Yu Cheng and Joakim Kjellsson for their very helpful comments.

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

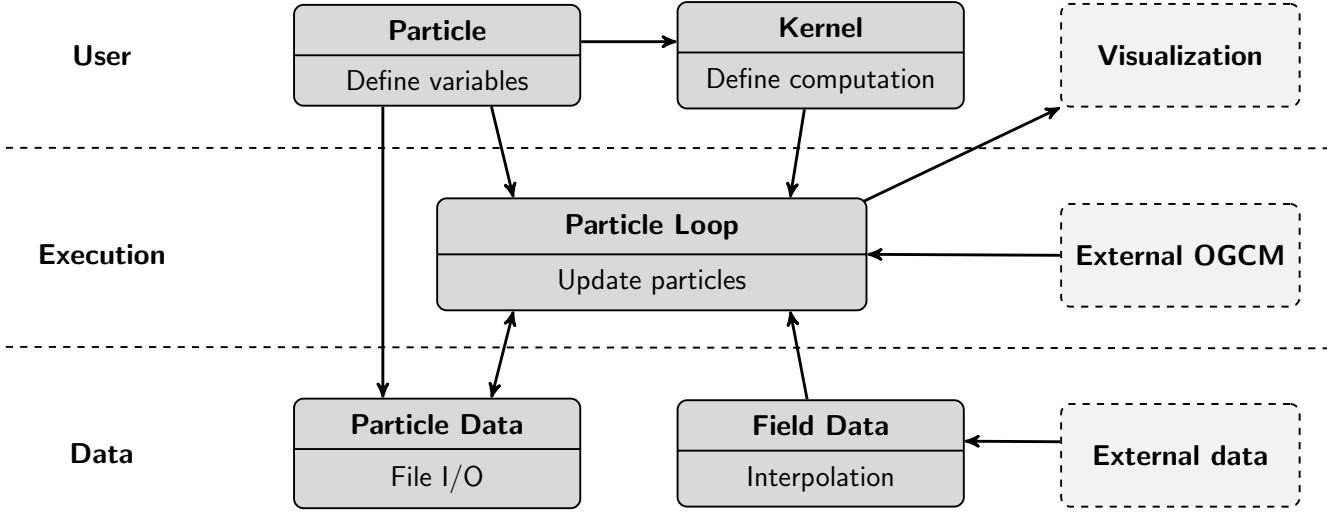

**Figure 1.** Conceptual abstractions (dark) and functionalities encapsulated in the Parcels prototype in relation to external components (light).

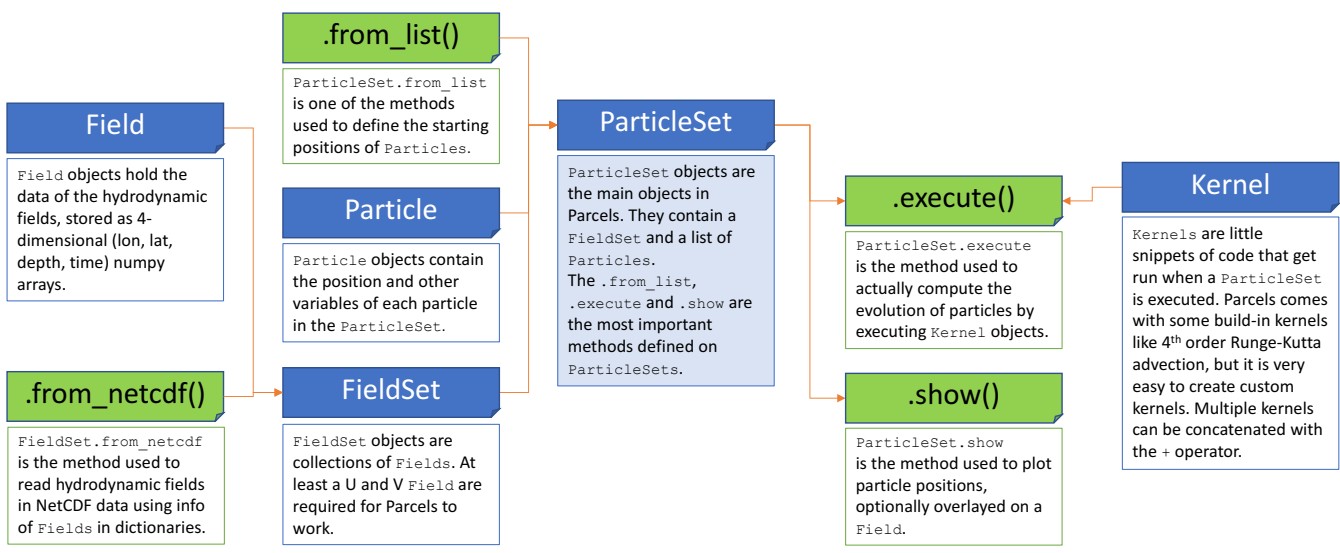

**Figure 2.** Class diagram of the Parcels v0.9 prototype implementation. Classes are depicted in blue, methods in green. Note that not all methods and classes are shown in this diagram.

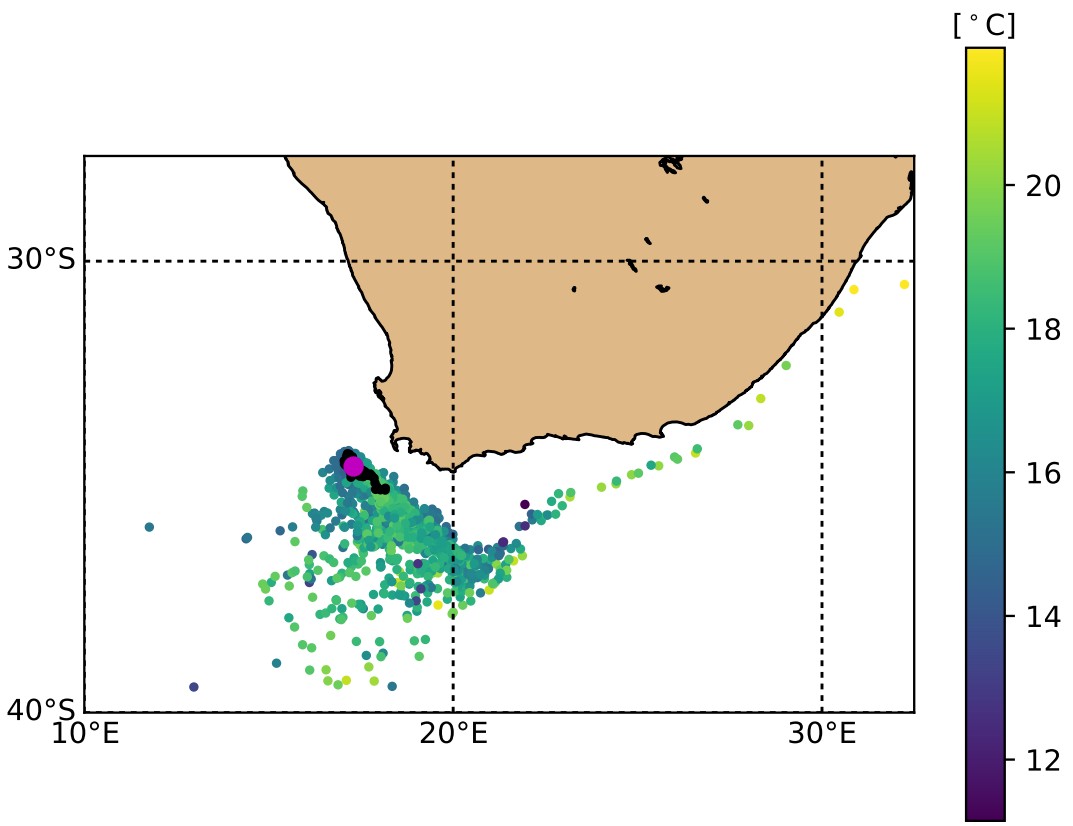

**Figure 3.** Footprints of virtual foraminifera ending up on the ocean floor just off Cape Town in the Agulhas region. This experiment is a Parcels implementation of the study described in Van Sebille et al. (2015), and this figure can be compared to Fig. 1b in that paper. The magenta dot is the location of the sediment core, from which virtual particles are first tracked back until they reach their 50m dwelling depth (black dots), and then further tracked back for their 30-day lifespan. Temperatures (in degrees Celcius) are recorded each day throughout their lifespan and shown as colours. The code for this experiment and plotting is available at https://doi.org/10.5281/zenodo.823994.

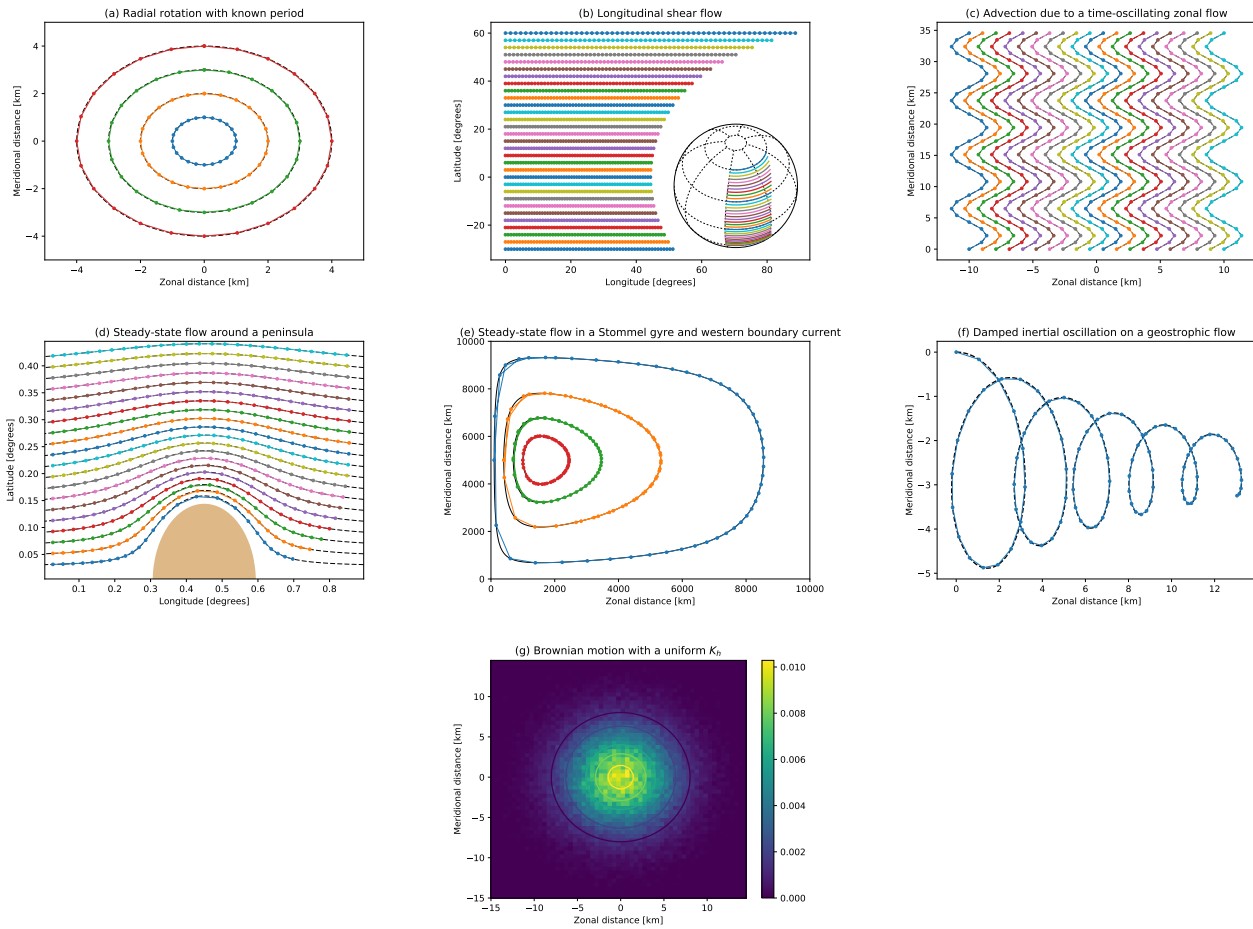

**Figure 4.** Evaluation of trajectory accuracy in Parcels v0.9, following the seven idealised and analytical test case described in in Sect. 6 of Van Sebille et al. (submitted): (a) radial rotation with known period; (b) longitudinal shear flow; (c) advection due to a time-oscillating zonal flow; (d) steady-state flow around a peninsula; (e) steady-state flow in a Stommel gyre and western boundary current; (f) damped inertial oscillation on a geostrophic flow; and (g) Brownian motion with a uniform $K_h$. In the upper six panels, the coloured lines are the particle trajectories and the black dashed lines are the analytical solutions. In panel (g), the colouring shows the density of particles, and the contours show the probability density function of the equivalent analytical solution (a two-dimensional Gaussian).