# Peer review of "Parcels v0.9: prototyping a Lagrangian Ocean Analysis framework for the petascale age"

_Geoscientific Model Development, 2017_

## Referee Comment (RC1) · J. Kjellsson (Referee) · 19 Aug 2017

The manuscript presents a new framework for a Lagrangian particle model, Parcels. The new particle model is in a testing phase with only the most basic components functional. The authors describe the current workings of the model, test its accuracy, and present where they envision development going further. Overall, the manuscript is well written. The main novelty lies in presenting the new framework to the particle-modelling community and its possible future developments. However, there are very few actual results. I recommend the paper for publication, but only after addressing the comments below.

[Figure]

Major comments

There is not much discussion about how exactly the new model will be more suited to cope with petascale age computing. The authors spend some time talking about how to optimise the loop over particles to improve performance, but with petascale OGCMs, where velocity fields amount to hundreds or more terabytes, reading and interpolating those fields into the particle model will be a huge bottleneck. Section 5.1 has a paragraph on how reading data from massive files could work, but there is no demonstration. In the practical example, the data file is 6Gb, which is not very large. I understand the authors have not focused on optimisation of PARCELS yet, but are there any examples, not necessarily with particle codes, where spatial indexing has given a performance improvement? I strongly recommend more discussion (in the introduction, design, and discussion sections) about how all current particle codes, e.g. CMS, Tracmass, Ariane, will hit this bottleneck in the peta-scale age, and more details about how PARCELS will overcome it.

Since PARCELS is very flexible, could it be extended to work for atmospheric particles? Perhaps PARCELS should not be presented as a tool for Lagrangian ocean analysis, but rather Lagrangian particle tracking in both atmosphere and oceans? Presenting this kind of framework to the atmospheric modelling community as well could be beneficial, but would mean changing the paper quite a bit. Even if the authors decide to stick with presenting PARCELS as an ocean particle code, atmospheric particle codes still need some mentioning (MetOffice NAME model, FLEXPART) in the introduction.

Minor comments

Throughout the paper, the authors name the model "Parcels". However, more than once I found that the name of the model could be confused with actual parcels. Why not use PARCELS, as any other model (e.g. NEMO, CESM, IFS etc.) to avoid confusion?

Page 1, Line 1: "petascale age" is rather unspecific. The sentence uses future tense, suggesting we are not there yet, even though there are already $> 1 peta flops$ computers. Please specify what is meant. OGCMs of a certain horizontal resolution, e.g. global $\sim 1/50$ or $\sim 1/100$?

Page1, Line 19: Add reference for seawater parcels: Doos 1995, Blanke & Raynaud 1997.

Page 2, Line 16: How would it keep up? By being very efficient at reading in velocity data?

Page 2, Line 18: I recommend replacing "functionality such as a myriad of behaviours to the particles" for "active particle behaviours".

Section 2.2: I found the section a bit confusing, and I think it needs some rewriting to become clearer. As I understood it, these are two methods for interpolating data, e.g. velocity, onto the particle position? I'm familiar with the interpolator from SciPy, but what method does the JIT method use? Is that something the user can write himself/herself? Is the SciPy interpolator restricted to nearest-neighbour or linear interpolation methods? Also, what pre-defined macros are you referring to?

Page 6, Line 11-13: Non-compatibility with non-regular grids excludes quite a few OGCMs, which often use rotated pole, tri-polar or cubed-sphere grids. I think this is one of the most important shortcomings of v0.9 that must be addressed soon by the authors or the user community. The authors should say so.

Page 7, Line 5-7: This sentence does not read well with its two parenthesis. I would split into two sentences, e.g. "The bash script get$_o fesdata_a gulhas.shprovidedathttps :$
$//doi.org/10.5281/zenodowasusedtodownloadsnapshots3165to3289, coveringtheyear2006, of a subdomain around the core si$

Page 8, Line 13: 6Gb is actually not a very large file. Many laptops have 16Gb RAM these days and could definitely cope with this while the user sips his/her coffee at some hip cafe.

Section 3.5: Does PARCELS write CF 1.6 compliant data?

Sections 4.2.1 - 4.2.7: The test cases need to be described a bit more. Are the fields generated within PARCELS, or generated and stored as netCDF files and then read into PARCELS? Also, please give $\Delta x, \Delta y, \Delta t$ for all fields.

Page 10, Line 28: "steady-state"

Section 5.1: Are there any tests that show that the optimisations they propose would give some performance improvement? Optimising the reading of velocity fields from very large files would be one of the main strengths of this model. See major comment above.

---

## Referee Comment (RC2) · Y. Cheng (Referee) · 24 Aug 2017

The authors introduce a new Lagrangian Ocean Analysis framework, Parcels that is built with flexibility, scalability, and ease-of-use in mind. This manuscript serves well as a proof-of-concept and demonstrates its user-friendly interfaces and the accuracy of the codes. However, the claimed improvement of performance, scalability for large datasets and ease to integrated with different OGCMs are yet to be seen.

The authors' effort to build Parcels following modern software engineering practice is much appreciated. From my experience, none of the existing tools, although gradually migrated to Github, use CI tools, making them difficult to be implemented to and tested

on different machines. This could be one of the reasons that each of community only reaches a certain size. Positioning itself as a framework, together with its modular design, Parcels has much more potentials than any existing tools to attract more users from different fields. At its early stage of development, Parcels urgently needs the input from a larger community, and this paper is a well-constructed invitation. I recommend the paper to be published once some of following points are addressed. Also, some of my questions and comments are also listed:

- P.2 line 6-8: Is there a limit that you expect each of three codes will not be computationally efficient anymore? From my own experience, instead of computation, the bottle neck of CMS is in the "output to NetCDF" step. Opening a huge NetCDF file and dumping particle information into it drain the system memory (32G/16core node) and dragged down the whole machine.

- P.2 line 28: It is not clear to me what "generic particle-mesh interaction computations." is.

- P.5 line 29: Is this limited to a particular JIT package? How much faster is using JITParticle than using ScipyParticle? It'd be good to see a benchmark comparison. If I want to use a new Interpolation scheme, how can I implement it into both modes?

- P.8 line 15 and the codes. Please correct me if my understanding is incorrect: The "snapshots" was defined in the code to load the data of initial three time steps. The for-loop was so designed because the particles are back tracked. The pset.execute uses the loaded filedset to advect particles in these three days (runtime) with time step (5min) and output every one day, and then "fldset.advancetime" is called to replace the last snapshot by concatenating the newly loaded snapshot to the front.

- P.7 line 8, and P.8 line 15: It is confusing to me that "Snapshots" was first a list of three members. Within the loop, it got updated to a single member list, which is the condition to trigger advancetime. Also, it might be better to point out that the "set_ofes_fieldset" is a user-defined function, not included in the Parcels package.

- Is there a particular reason that field.py uses the native Netcdf4 API to load in fields, but uses Xarray to write to output files? Performance concerned?

- Section 4.2: How did you generate the idealized flow fields? It might be trivial, but the grid-size information is missing for some cases.

- Section 4.2.7: I am just curious why $K_{h}=100 \ m^{2}/s$? From my experience, this value depends on resolution. Let's say if I want to use Parcels with 1/10 deg OFES, what value should I use? Is there a test case that I can tune this value against some observations?

- P.12 line 16: Not exactly clear what "Spatial indexing methods" indicates. Also, this sentence sounds not clear to me: How will something "impacted" become promising in the future? How are you planning to do the "close integration of such scheduling techniques with the particle loop?"

- Section 5.1.1: How are you planning to proceed on this front? I know there is a "Parallel NetCDF" project, but I don't think its Python API will be available anytime soon. How much of performance gain could be achieved once the second point is addressed in version 1.0?

- Section 5.3: This is exciting! It would be wonderful to be able to couple Parcels to many ocean models. However, to do so, Parcels need to work on their native grids. It is possible to run a remapping layer in between, but I think that will significantly impact the performance. Just curious, could you briefly suggest how to modify the field.py to handle the non-regular grids, for example, the tri-polar grid in POP2, or the unstructured triangular meshes in FESOM? What's your current plan to support multiple grids? Or will that be left to the users?
* * *

---

## Author Comment (AC1) · 26 Sep 2017

Response to Reviewer 1: Dr Joakim Kjellsson

*The manuscript presents a new framework for a Lagrangian particle model, Parcels. The new particle model is in a testing phase with only the most basic components functional. The authors describe the current workings of the model, test its accuracy, and present where they envision development going further. Overall, the manuscript is well written. The main novelty lies in presenting the new framework to the particle-modelling community and its possible future developments. However, there are very few actual results. I recommend the paper for publication, but only after addressing the*

[Figure]

*comments below.*

We thank Dr Kjellsson for these kind words, and have indeed addressed all his comments, as detailed below

*There is not much discussion about how exactly the new model will be more suited to cope with petascale age computing. The authors spend some time talking about how to optimise the loop over particles to improve performance, but with petascale OGCMs, where velocity fields amount to hundreds or more terabytes, reading and interpolating those fields into the particle model will be a huge bottleneck. Section 5.1 has a paragraph on how reading data from massive files could work, but there is no demonstration. In the practical example, the data file is 6Gb, which is not very large. I understand the authors have not focused on optimisation of PARCELS yet, but are there any examples, not necessarily with particle codes, where spatial indexing has given a performance improvement? I strongly recommend more discussion (in the introduction, design, and discussion sections) about how all current particle codes, e.g. CMS, Tracmass, Ariane, will hit this bottleneck in the peta-scale age, and more details about how PARCELS will overcome it.*

We agree that the future performance challenges have not been addressed in sufficient detail and have made additions to the suggested sections 1 (p 2 of the track-changed pdf, lines 9-12), 2.2 and the new 2.3 (p 6 of the track-changed pdf, lines 18-32 and page 7 of the track-changed pdf, lines 1-4), 3.1 (p 8 of the track-changed pdf, lines 5-8) and 5.1 (p 14 of the track-changed pdf, lines 1-20).

The overall rationale follows the argument that, as no "silver bullet" solution is as yet known for the big-data challenges facing Lagrangian tracking applications with petascale OGCMs, we see the primary limitations of existing codes in the lack of flexibility and dynamic adaptability that is required to explore new optimisation strategies. Throughout the listed sections we explore two potential solutions for dealing with the vast volumes of field data required, for two different usage scenarios of Lagrangian

tracking models: a) Coupled execution with the host model to avoid the bandwidth limitations of disk and file storage; and b) reducing the required data volume by selectively prefetching field data based on known particle positions. We aim to highlight throughout that both approaches have been considered during the design of Parcels to enable efficient exploration of such schemes during future development.

*Since PARCELS is very flexible, could it be extended to work for atmospheric particles? Perhaps PARCELS should not be presented as a tool for Lagrangian ocean analysis, but rather Lagrangian particle tracking in both atmosphere and oceans? Presenting this kind of framework to the atmospheric modelling community as well could be beneficial, but would mean changing the paper quite a bit. Even if the authors decide to stick with presenting PARCELS as an ocean particle code, atmospheric particle codes still need some mentioning (MetOffice NAME model, FLEXPART) in the introduction.*

This is an interesting suggestion. While our own development for now focusses on oceanographic applications, the framework could indeed in principle also be used in the atmosphere. We have now added some discussion of atmospheric particle tracking codes in the introduction (p 2 of the track-changed pdf, lines 27-30).

*Throughout the paper, the authors name the model "Parcels". However, more than once I found that the name of the model could be confused with actual parcels. Why not use PARCELS, as any other model (e.g. NEMO, CESM, IFS etc.) to avoid confusion?*

We thank the reviewer for this thoughtful suggestion, but we feel that the current convention is more concise and in the spirit of the Python philosophy. The suggested all-caps naming style is largely derived from Fortran90 coding conventions, where routine and package names are often capitalised, while it is preferred in the Python community to use lower-case module names (https://www.python.org/dev/peps/pep-0008/package-and-module-names). Since this is largely a matter of personal taste, we feel that we must respectfully decline.

*Page 1, Line 1: "petascale age" is rather unspecific. The sentence uses future tense,*

*suggesting we are not there yet, even though there are already > 1petaflops computers. Please specify what is meant. OGCMs of a certain horizontal resolution, e.g. global $\equiv$ 1/50 or $\equiv$ 1/100?*

We have now clarified in the abstract that by petascale age we mean output of OCGCM models exceeding the 1 petabyte storage space barrier (p 1 of the track-changed pdf, lines 2-3). This would for example be the case for a global simulation at 1/50o run for 50 years and stored at daily intervals.

*Page1, Line 19: Add reference for seawater parcels: Doos 1995, Blanke Raynaud 1997.*

We have now added references to Döös 1995 and Blanke Raynaud 1997 to this sentence (p 1 of the track-changed pdf, line 19)

*Page 2, Line 16: How would it keep up? By being very efficient at reading in velocity data?*

We have now clarified that indeed we see scalability and efficiency in reading in hydrodynamic data as key strategies to keep up with OGCM development (p 2 of the track-changed pdf, lines 21).

*Page 2, Line 18: I recommend replacing "functionality such as a myriad of behaviours to the particles" for "active particle behaviours".*

We have changed this phrasing following the reviewer's suggestion (p 2 of the track-changed pdf, line 23)

*Section 2.2: I found the section a bit confusing, and I think it needs some rewriting to become clearer. As I understood it, these are two methods for interpolating data, e.g. velocity, onto the particle position? I'm familiar with the interpolator from SciPy, but what method does the JIT method use? Is that something the user can write himself/herself? Is the SciPy interpolator restricted to nearest-neighbour or linear interpolation methods? Also, what pre-defined macros are you referring to?*

To make this clearer we have separate the details of the provided interpolation routines from the overview of the execution loop and added a new subsection 2.3 (p 6/7 of the track-changed pdf, lines 23-4). This now clearly states the current implementation details, its restrictions and potential for additional interpolation schemes in the future.

*Page 6, Line 11-13: Non-compatibility with non-regular grids excludes quite a few OGCMs, which often use rotated pole, tri-polar or cubed-sphere grids. I think this is one of the most important shortcomings of v0.9 that must be addressed soon by the authors or the user community. The authors should say so.*

We fully agree with the reviewer, and indeed extension to curvilinear grid is the next major development goal for Parcels. We also agree that we should state this major limitation more upfront, and have now mentioned it in the abstract too (p 1 of the track-changed pdf, line 15).

*Page 7, Line 5-7: This sentence does not read well with its two parenthesis. I would split into two sentences*

We have changed the phrasing following the reviewer's suggestion (p 8 of the track-changed pdf, lines2-4).

*Page 8, Line 13: 6Gb is actually not a very large file. Many laptops have 16Gb RAM these days and could definitely cope with this while the user sips his/her coffee at some hip cafe.*

We have limited the data set here to 6GB in order to keep the downloading of the data manageable, for users who want to run this experiment themselves. We thought some users might find it problematic to be asked to download 100s of GB. This example is meant to highlight the API, not per se to show the scalability of the code (which will be optimised in future version of Parcels anyways). The 6 GB of data was hence a compromise between 'manageable' download volume and sufficient data to conduct a meaningful experiment.

*Section 3.5: Does PARCELS write CF 1.6 compliant data?*

The output of Parcels is indeed CF-1.6 compliant, following the guide-lines at http://cfconventions.org/cf-conventions/v1.6.0/cf-conventions.html#discrete-sampling-geometries. We have now explicitly mentioned that at the beginning of section 3.5 (p 10 of the track-changed pdf, lines 2-4).

*Sections 4.2.1 - 4.2.7: The test cases need to be described a bit more. Are the fields generated within PARCELS, or generated and stored as netCDF files and then read into PARCELS? Also, please give $\triangle x$, $\triangle y$, $\triangle t$ for all fields.*

The reviewer was right that not all of the test cases provided sufficient information on how the fields were generated, and at what resolution. We have now added that information (p 11 of the track-changed pdf, lines 8-10, 12-15 and 23; p 12 of the track-changed pdf, lines 8-9, 18-19 and 29; and p 13 of the track-changed pdf, line 12).

*Page 10, Line 28: "steady-state"*

We have fixed the type-o (p 12 of the track-changed pdf, line 15)

*Section 5.1: Are there any tests that show that the optimisations they propose would give some performance improvement? Optimising the reading of velocity fields from very large files would be one of the main strengths of this model. See major comment above.*

Unfortunately, there is no hard evidence or citable publications for any of the proposed methods yet, which is why further experimentation is required. The general concepts of out-of-core streaming for large data-sets have been very skillfully addressed by a combination of two prominent Python packages recently, namely xarray and dask. This method significantly reduces the memory overhead of large files, and could possibly be extended to incorporate particle-specific information to further optimise the file reads, as outlined in the re-written parts of section 5.1 (page 14 of the track-changed pdf). The flexible design structure and native compatibility with these Python packages

will allow for efficient exploration of such advanced methods in future work.

Please also note the supplement to this comment:
https://www.geosci-model-dev-discuss.net/gmd-2017-167/gmd-2017-167-AC1-
supplement.pdf

—————————————————

[Figure]

**Supplement:**

[revised manuscript text omitted]

---

## Author Comment (AC2) · 26 Sep 2017

Reviewer 2: Dr Yu Cheng

*The authors introduce a new Lagrangian Ocean Analysis framework, Parcels that is built with flexibility, scalability, and ease-of-use in mind. This manuscript serves well as a proof-of-concept and demonstrates its user-friendly interfaces and the accuracy of the codes. However, the claimed improvement of performance, scalability for large datasets and ease to integrated with different OGCMs are yet to be seen.*

*The authors' effort to build Parcels following modern software engineering practice is*

[Figure]

*much appreciated. From my experience, none of the existing tools, although gradually migrated to Github, use CI tools, making them difficult to be implemented to and tested on different machines. This could be one of the reasons that each of community only reaches a certain size. Positioning itself as a framework, together with its modular design, Parcels has much more potentials than any existing tools to attract more users from different fields. At its early stage of development, Parcels urgently needs the input from a larger community, and this paper is a well-constructed invitation. I recommend the paper to be published once some of following points are addressed. Also, some of my questions and comments are also listed:*

We thank Dr Cheng for these very kind words. Below we have listed our responses to his comments

*P.2 line 6-8: Is there a limit that you expect each of three codes will not be computationally efficient anymore? From my own experience, instead of computation, the bottle neck of CMS is in the "output to NetCDF" step. Opening a huge NetCDF file and dumping particle information into it drain the system memory (32G/16core node) and dragged down the whole machine.*

This is a very good point. While it is beyond the scope of this paper to speculate when the other code to not be computationally efficient anymore (that also depends on development by their teams), we do agree that particle trajectory writing can be a major bottleneck too. We have now highlighted this in the introduction (p 2 of the track-changed pdf, lines 9).

*P.2 line 28: It is not clear to me what "generic particle-mesh interaction computations." is.*

The sentence has been rewritten to make the intention clearer (p 3 of the track-changed pdf, lines 3-4).

*P.5 line 29: Is this limited to a particular JIT package? How much faster is using*

*JITParticle than using ScipyParticle? It'd be good to see a benchmark comparison. If I want to use a new Interpolation scheme, how can I implement it into both modes?*

The JIT components used by Parcels are tightly integrated into the code generation engine and are internal to Parcels. The primary advantage of using code generation and JIT processing is that the low-level interpolation routines can be tightly coupled with the processing kernels via inlining, which is orders of magnitude faster than the pure Python mode, due to the large overhead of calling native Python functions repeatedly in tight loops. This is now explained in more detail in the newly added section 2.3 (p 6 and 7 of the track-changed pdf, lines 23-4), and an "anecdotal" performance benchmark has been added to illustrate the performance improvement due to JIT mode.

*P.8 line 15 and the codes. Please correct me if my understanding is incorrect: The "snapshots" was defined in the code to load the data of initial three time steps. The for-loop was so designed because the particles are back tracked. The pset.execute uses the loaded filedset to advect particles in these three days (runtime) with time step (5min) and output every one day, and then "fldset.advancetime" is called to replace the last snapshot by concatenating the newly loaded snapshot to the front.*

This is indeed correct; we have now further clarified the use of the advancetime function in sections 3.1 and 3.4 (p 9 of the track-changed pdf, lines 19-21).

*P.7 line 8, and P.8 line 15: It is confusing to me that "Snapshots" was first a list of three members. Within the loop, it got updated to a single member list, which is the condition to trigger advancetime. Also, it might be better to point out that the "set_ofes_fieldset" is a user-defined function, not included in the Parcels package.*

In the revised version of the manuscript, we now explicitly state that set_ofes_fieldset is indeed a user-defined function (p 8 of the track-changed pdf, lines 9-11). We have also clarified that this function accepts a list of any length.

*Is there a particular reason that field.py uses the native Netcdf4 API to load in fields,*

*but uses Xarray to write to output files? Performance concerned?*

The use of the native netCDF API is largely an overhead from earlier development stages and will be addressed in future releases. The plan is to incorporate Xarray with the scheduling library Dask to facilitate efficient out-of-core file reads that complement the optimised particle computation. The primary work on this is planned for the upcoming performance optimization and parallelization phases.

*Section 4.2: How did you generate the idealized flow fields? It might be trivial, but the grid-size information is missing for some cases.*

The reviewer was right that not all of the test cases provided sufficient information on how the fields were generated, and at what resolution. We have now added that information (p 11 of the track-changed pdf, lines 8-10, 12-15 and 23; p 12 of the track-changed pdf, lines 8-9, 18-19 and 29; and p 13 of the track-changed pdf, line 12).

*Section 4.2.7: I am just curious why $K_h = 100m^2/s$? From my experience, this value depends on resolution. Let's say if I want to use Parcels with 1/10 deg OFES, what value should I use? Is there a test case that I can tune this value against some observations?*

The $K_h = 100m^2/s$ here was purely chosen as an illustrative example. The reviewer raises a very good question about appropriate values for $K_h$, but we feel that (or even if Brownian motion is a good model of diffusion in the ocean anyway) is beyond the scope of the paper here. We know that quite a few groups, including for example Arne Biastoch's in Kiel, are actively investigating the question raised here.

*P.12 line 16: Not exactly clear what "Spatial indexing methods" indicates. Also, this sentence sounds not clear to me: How will something "impacted" become promising in the future? How are you planning to do the "close integration of such scheduling techniques with the particle loop?"*

Section 5.1 has been reworded to explain potential performance optimizations in more

detail, including the use of spatial indexing methods to reduce the overall field data volume. Additional references have also been added for background on geospatial indexing (p 13 of the track-changed pd).

*Section 5.1.1: How are you planning to proceed on this front? I know there is a "Parallel NetCDF" project, but I don't think its Python API will be available anytime soon. How much of performance gain could be achieved once the second point is addressed in version 1.0?*

Various projects exist that improve on the current shortcomings of the netCDF Python wrappers. The most popular and established package is probably Xarray, which provides parallelisation (via Dask) and out-of-core computation, and could potentially be used to implement grid parallelisation based on spatial decomposition methods.

*Section 5.3: This is exciting! It would be wonderful to be able to couple Parcels to many ocean models. However, to do so, Parcels need to work on their native grids. It is possible to run a remapping layer in between, but I think that will significantly impact the performance. Just curious, could you briefly suggest how to modify the field.py to handle the non-regular grids, for example, the tri-polar grid in POP2, or the unstructured triangular meshes in FESOM? What's your current plan to support multiple grids? Or will that be left to the users?*

The newly added section 2.3 adds a few details about the current assumptions made in the code about field interpolation, and its potential for future extensions (p 6 and 7 of the track-changed pdf, lines 23-24). The primary idea is that the current Field class acts as an abstract base class for future field types with individual field-specific interpolation routines provided as macros to the code-generation engine to include at runtime. While this will require developer input and expertise, such additions would be relatively concise and should not affect existing user models, allowing a relatively easy transition between offline and online models.

Please also note the supplement to this comment:
https://www.geosci-model-dev-discuss.net/gmd-2017-167/gmd-2017-167-AC2-supplement.pdf
* * *
[Figure]

**Supplement:**

[revised manuscript text omitted]